# Revisiting the Train Loss: an Efficient Performance Estimator for Neural Architecture Search

## Abstract

Reliable yet efficient evaluation of generalisation performance of a proposed architecture is crucial to the success of neural architecture search (NAS). Traditional approaches face a variety of limitations: training each architecture to completion is prohibitively expensive, early stopping estimates may correlate poorly with fully trained performance, and model-based estimators require large training sets. Instead, motivated by recent results linking training speed and generalisation with stochastic gradient descent, we propose to estimate the final test performance based on the sum of training losses. Our estimator is inspired by the marginal likelihood, which is used for Bayesian model selection. Our model-free estimator is simple, efficient, and cheap to implement, and does not require hyperparameter-tuning or surrogate training before deployment. We demonstrate empirically that our estimator consistently outperforms other baselines under various settings and can achieve a rank correlation of 0.95 with final test accuracy on the NAS-Bench201 dataset within 50 epochs.

## 1 Introduction

Reliably estimating the generalisation performance of a proposed architecture is crucial to the success of Neural Architecture Search (NAS) but has always been a major bottleneck in NAS algorithms (Elsken et al., 2018). The traditional approach of training each architecture for a large number of epochs and evaluating it on validation data (*full evaluation*) provides a reliable performance measure, but requires prohibitively high computational resources on the order of thousands of GPU days (Zoph & Le, 2017; Real et al., 2017; Zoph et al., 2018; Real et al., 2019; Elsken et al., 2018). This motivates the development of methods for speeding up performance estimation to make NAS practical for limited computing budgets. A popular simple approach is *early-stopping* which offers a low-fidelity approximation of generalisation performance by training for fewer epochs (Li et al., 2016; Falkner et al., 2018; Li & Talwalkar, 2019). However, if we stop the training early at a small number of epochs and evaluate the model on validation data, the relative performance ranking may not correlate well with the performance ranking of the full evaluation (Zela et al., 2018). Another line of work focuses on *learning curve extrapolation* (Domhan et al., 2015; Klein et al., 2016b; Baker et al., 2017), which trains a surrogate model to predict the final generalisation performance based on the initial learning curve and/or meta-features of the architecture. However, the training of the surrogate often requires hundreds of fully evaluated architectures to achieve satisfactory extrapolation performance and the hyper-parameters of the surrogate also need to be optimised. Alternatively, the idea of *weight sharing* is adopted in one-shot NAS methods to speed up evaluation (Pham et al., 2018; Liu et al., 2019; Xie et al., 2019b). Despite leading to significant cost-saving, weight sharing heavily underestimates the true performance of good architectures and is unreliable in predicting the relative ranking among architectures (Yang et al., 2020; Yu et al., 2020).

In view of the above limitations, we propose a simple model-free method which provides a reliable yet computationally cheap estimation of the generalisation performance ranking of architectures: the Sum over Training Losses (SoTL). Our method harnesses the training losses of the commonly-used SGD optimiser during training, and is motivated by recent empirical and theoretical results linking training speed and generalisation (Hardt et al., 2016; Lyle et al., 2020). We ground our method in the Bayesian update setting, where we show that the SoTL estimator computes a lower bound to the

model evidence, a quantity with sound theoretical justification for model selection (MacKay, 1992). We show empirically that our estimator can outperform a number of strong existing approaches to predict the relative performance ranking among architectures, while speeding up different NAS approaches significantly.

## 2 METHOD

We propose a simple metric that estimates the generalisation performance of a deep neural network model via the Sum of its Training Losses (SoTL). After training a deep neural network whose prediction is $f_\theta(\cdot)$ for $T$ epochs[1], we sum the training losses collected so far:

$$SoTL = \sum_{t=1}^{T} \left[ \frac{1}{B} \sum_{i=1}^{B} l\left(f_{\theta_{t,i}}(\mathbf{X}_i), \mathbf{y}_i\right) \right] \tag{1}$$

where $l$ is the training loss of a mini-batch $(\mathbf{X}_i, \mathbf{y}_i)$ at epoch $t$ and $B$ is the number of training steps within an epoch. If we use the first few epochs as the burn-in phase for $\theta_{t,i}$ to converge to certain distribution $P(\theta)$ and start the sum from epoch $t = T - E + 1$ instead of $t = 1$, we obtain a variant SoTL-E. In the case where $E = 1$, we start the sum at $t = T$ and our estimator corresponds to the sum over training losses within epoch $t = T$. We discuss SoTL's theoretical interpretation based on Bayesian marginal likelihood and training speed in Section 3, and empirically show that SoTL, despite its simple form, can reliably estimate the generalisation performance of neural architectures in Section 5.

If the sum over training losses is a useful indicator for the generalisation performance, one might expect the sum over validation losses to be a similarly effective performance estimator. The sum over validation losses (SoVL) lacks the link to the Bayesian model evidence, and so its theoretical motivation is different from our SoTL. Instead, the validation loss sum can be viewed as performing a bias-variance trade-off; the parameters at epoch $t$ can be viewed as a potentially high-variance sample from a noisy SGD trajectory, and so summation reduces the resulting variance in the validation loss estimate at the expense of incorporating some bias due to the relative ranking of models' test performance changing during training. We show in Section 5 that SoTL clearly outperforms SoVL in estimating the true test performance.

## 3 THEORETICAL MOTIVATION

The SoTL metric is a direct measure of training speed and draws inspiration from two lines of work: the first is a Bayesian perspective that connects training speed with the marginal likelihood in the model selection setting, and the second is the link between training speed and generalisation (Hardt et al., 2016). In this section, we will summarize recent results that demonstrate the connection between SoTL and generalisation, and further show that in Bayesian updating regimes, the SoTL metric corresponds to an estimate of a lower bound on the model's marginal likelihood, under certain assumptions.

### 3.1 TRAINING SPEED AND THE MARGINAL LIKELIHOOD

We motivate the SoTL estimator by a connection to the model evidence, also called the marginal likelihood, which is the basis for Bayesian model selection. The model evidence quantifies how likely a dataset $\mathcal{D}$ is to have been generated by a model, and so can be used to update a prior belief distribution over which model from a given set is most likely to have generated $\mathcal{D}$. Given a model with parameters $\theta$, prior $\pi(\theta)$, and likelihood $P(\mathcal{D}|\theta)$ for a training data set $\mathcal{D} = \{\mathcal{D}_1, \ldots, \mathcal{D}_n\}$ with data points $\mathcal{D}_i = (x_i, y_i)$, the (log) marginal likelihood is expressed as follows.

$$\log P(\mathcal{D}) = \log \mathbb{E}_{\pi(\theta)}\left[P(\mathcal{D}|\theta)\right] \Leftrightarrow \log P(\mathcal{D}) = \sum_{i=1}^{n} \log P(\mathcal{D}_i|\mathcal{D}_{<i}) = \sum_{i=1}^{n} \log\left[\mathbb{E}_{P(\theta|\mathcal{D}_{<i})}\left[P(\mathcal{D}_i|\theta)\right]\right]$$

Interpreting the negative log posterior predictive probability $-\log P(\mathcal{D}_i|\mathcal{D}_{<i})$ of each data point as a 'loss' function, the log evidence then corresponds to the area under a training loss curve, where each

---

[1]$T$ can be far from the total training epochs $T_{end}$ used in complete training

training step would be computed by sampling a data point $\mathcal{D}_i$, taking the log expected likelihood under the current posterior $P(\theta|\mathcal{D}_{<i})$ as the current loss, and then updating the posterior by incorporating the new sampled data point: $\mathcal{D}_{<i+1} := \mathcal{D}_{<i} \cup \{\mathcal{D}_i\}$. One can therefore interpret the marginal likelihood as a measure of training speed in a Bayesian updating procedure. In the setting where we cannot compute the posterior analytically and only samples $\hat{\theta}$ from the posterior over parameters are available, we obtain an unbiased estimator of a lower bound $\mathcal{L}(\mathcal{D}) = \sum \mathbb{E}_{P(\theta|\mathcal{D}_{<i})} [\log P(\mathcal{D}_i|\theta)]$ on the marginal likelihood by Jensen's inequality, which again corresponds to minimizing a sum over training losses

$$\sum \log P(\mathcal{D}_i|\hat{\theta}) \approx \sum \mathbb{E}_{P(\theta|\mathcal{D}_{<i})} [\log P(\mathcal{D}_i|\theta)] \leq \sum \log \left[ \mathbb{E}_{P(\theta|\mathcal{D}_{<i})}[P(\mathcal{D}_i|\theta)] \right] = \log P(\mathcal{D})$$

with $\approx$ denoting equality in expectation. A full analysis of the Bayesian setting is outside of the scope of this work. We refer the reader to (Lyle et al., 2020) for more details of the properties of this estimator in Bayesian models. Although the NAS setting does not yield the same interpretation of SoTL as model evidence estimation, we argue that the SoTL metric is still *plausibly* useful for model selection. Just as the marginal likelihood measures the utility of updates based on early data points in predicting later data points, the SoTL of a model trained with SGD will be lower for models whose mini-batch gradient descent updates improve the loss of later mini-batches seen during optimisation. We refer the reader to Apppendix B to see a demonstration of the SoTL metric in the Bayesian linear regression setting. We emphasize that the Bayesian connection thus justifies the *sum* over training losses as a tool for model selection, but not the training loss from a single parameter update.

### 3.2 TRAINING SPEED AND GENERALISATION

Independent of the accuracy of SoTL in estimating the Bayesian model evidence, it is also possible to motivate our method by its relationship with training speed: models which achieve low training loss quickly will have low SoTL. There are both empirical and theoretical lines of work that illustrate a deep connection between training speed and generalisation. On the theoretical front, we find that models which train quickly can attain lower generalisation bounds. Training speed and generalisation can be related via stability-based generalisation bounds (Hardt et al., 2016; Liu et al., 2017), which characterize the dependence of the solution found by a learning algorithm on its training data. In networks of sufficient width, (Arora et al., 2019) propose a neural-tangent-kernel-based data complexity measure which bounds both the convergence rate of SGD and the generalisation error of the model obtained by optimisation. A similar generalisation bound and complexity measure is obtained by (Cao & Gu, 2019).

While theoretical work has largely focused on ranking *bounds* on the test error, current results do not provide guarantees on consistency between the ranking of different models' test set performance and their generalisation bounds. The empirical work of (Jiang* et al., 2020) demonstrates that many complexity measures are uncorrelated or negatively correlated with the relative performance of models on their test data but notably, a particular measure of training speed – the number of steps required to reach cross-entropy loss of 0.1, was highly correlated with the test set performance ranking of different models. The connection between training speed and generalisation is also observed by (Zhang et al., 2016), who find that models trained on true labels converge faster than models trained on random labels, and attain better generalisation performance.

## 4 RELATED WORK

Various approaches have been developed to speed up architecture performance estimation, thus improving the efficiency of NAS. Low-fidelity estimation methods accelerate NAS by using the validation accuracy obtained after training architectures for fewer epochs (namely early-stopping) (Li et al., 2016; Falkner et al., 2018; Zoph et al., 2018; Zela et al., 2018), training a down-scaled model with fewer cells during the search phase (Zoph et al., 2018; Real et al., 2019) or training on a subset of the data (Klein et al., 2016a). However, low-fidelity estimates underestimate the true performance of the architecture and can change the relative ranking among architectures (Elsken et al., 2018). This undesirable effect on relative ranking is more prominent when the cheap approximation set-up is too dissimilar to the full evaluation (Zela et al., 2018). As shown in our Fig. 2 below, the validation accuracy at early epochs of training suffers low rank correlation with the final test performance.

Another way to cheaply estimate architecture performance is to train a regression model to extrapolate the learning curve from what is observed in the initial phase of training. Regression model choices that have been explored include Gaussian processes with a tailored kernel function (Domhan et al., 2015), an ensemble of parametric functions (Domhan et al., 2015), a Bayesian neural network (Klein et al., 2016b) and more recently a $\nu$-support vector machine regressor ($\nu$-SVR)(Baker et al., 2017) which achieves state-of-the-art prediction performance. Although these model-based methods can often predict the performance ranking better than their model-free early-stopping counterparts, they require a relatively large amount of fully evaluated architecture data (e.g. 100 fully evaluated architectures in (Baker et al., 2017)) to train the regression surrogate properly and optimise the model hyperparameters in order to achieve good prediction performance. The high computational cost of collecting the training set makes such model-based methods less favourable for NAS unless the practitioner has already evaluated hundreds of architectures on the target task. Moreover, both low-fidelity estimates and learning curve extrapolation estimators are empirically developed and lack theoretical motivation.

Finally, one-shot NAS methods employ weight sharing to reduce computational costs (Pham et al., 2018; Liu et al., 2019; Xie et al., 2019b). Under the one-shot setting, all architectures are considered as subgraphs of a supergraph. Only the weights of the supergraph are trained while the architectures (subgraphs) inherit the corresponding weights from the supergraph. Weight sharing removes the need for retraining each architecture during the search and thus achieves a significant speed-up. However, the weight sharing ranking among architectures often correlates very poorly with the true performance ranking (Yang et al., 2020; Yu et al., 2020; Zela et al., 2020), meaning architectures chosen by one-shot NAS are likely to be sub-optimal when evaluated independently (Zela et al., 2020). Moreover, one-shot methods are often outperformed by sample-based NAS methods (Dong & Yang, 2020; Zela et al., 2020).

Apart from the above mentioned performance estimators used in NAS, many complexity measures have been proposed to analyse the generalisation performance of deep neural networks. (Jiang* et al., 2020) provides a rigorous empirical analysis of over 40 such measures. This investigation finds that sharpness-based measures (McAllester, 1999; Keskar et al., 2016; Neyshabur et al., 2017; Dziugaite & Roy, 2017) (including PAC-Bayesian bounds) provide good correlation with test set performance, but their estimation requires adding randomly generated perturbations to the network parameters and the magnitude of the perturbations needs to be carefully optimised with additional training, making them unsuitable performance estimators for NAS. Optimisation-based complexity measures also perform well in predicting generalisation. Specifically, the number of steps required to reach loss of 0.1, as mentioned in Section 3.2, is closely related to our approach as both quantities measure the training speed of architectures. To our knowledge though, this measure has never been used in the NAS context before.

## 5 EXPERIMENTS

In this section we compare the following measures. Note $T$ denotes the intermediate training epoch, which is smaller than the final epoch number $T_{end} > T$: Our proposed estimator **Sum of training losses over all preceding epochs (SoTL)**, which sums the training losses of an architecture from epoch $t = 0$ to the current epoch $t = T$, and its variant **Sum of training losses over the most recent $E$ epochs (SoTL-E)**, which uses the sum of the training losses from epoch $t = T - E$ to $t = T$. **Sum of validation losses over all preceding epochs (SoVL)** computes the sum of the *validation* losses of an neural architecture from epoch $t = 0$ to the current epoch $t = T$. **Validation accuracy at an early epoch (VAccES)** corresponds to early-stopping practice whereby the user assumes the validation accuracy of an architecture at early epoch $t = T < T_{end}$ is a good estimator of its final test performance at epoch $t = T_{end}$. **Learning curve extrapolation (LcSVR)** method is the state-of-the-art extrapolation method proposed in (Baker et al., 2017) which uses a trained $\nu$-SVR to predict the final validation accuracy of an architecture. The inputs for the SVR regression model comprise architecture meta-features (e.g. number of parameters and depth of the architecture), training hyper-parameters (e.g. initial learning rate, mini-batch size and weight decay), learning curve features up to epoch $t = T$ (e.g. the validation accuracies up to epoch $t = T$, the 1st-order and 2nd-order differences of validation curve up to epoch $t = T$). In our experiments, we train the SVR on data of 200 randomly sampled architectures and following the practice in (Baker et al., 2017), we optimise the SVR hyperparameters via random search using 3-fold cross-validation. We also compare against

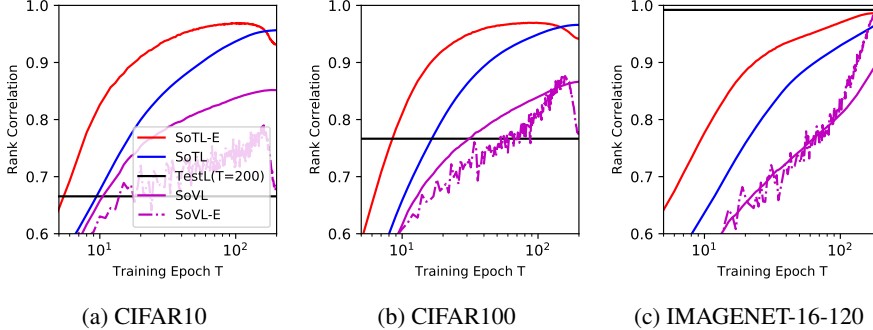

(a) CIFAR10       (b) CIFAR100       (c) IMAGENET-16-120

Figure 1: Rank correlation (with final test *accuracy*) performance of the sum of training losses, SoTL (blue) and SoTL-E (red), and those of validation losses (purple), SoVL (solid) and SoVL-E (dash dot), as well as that of final test loss (black) for 5000 random architectures in NASBench-201 on three image datasets. Note the correlation performances of final test loss and SoVL-E near the end of training get surprisingly poor for CIFAR10/100. We explain this in Section 5.1.

with two baselines on the DARTS search space: **the training losses at each mini batch (TLmini)** and the variant of VAccES, **VAccES(EMA)** whereby the exponential moving average of the weights (Tan & Le, 2019) is used during validation to improve validation accuracy.

The datasets we used to compare these performance estimators are:

- **NASBench-201** (Dong & Yang, 2020): the dataset contains information of 15,625 different neural architectures, each of which is trained with SGD optimiser for 200 epochs ($T_{end} = 200$) and evaluated on 3 different datasets: CIFAR10, CIFAR100, IMAGENET-16-120. The NASBench-201 datasets can be used to benchmark almost all up-to-date NAS search strategies.

- **RandWiredNN**: we produced this dataset by generating 552 randomly wired neural architectures from the random graph generators proposed in (Xie et al., 2019a) and evaluating the architecture performance on the FLOWERS102 dataset (Nilsback & Zisserman, 2008). We explored 69 sets of hyperparameter values for the random graph generators and for each set of hyperparameter values, we sampled 8 randomly wired neural networks from the generator. All the architectures are trained with SGD optimiser for 250 epochs ($T_{end} = 250$). This dataset allows us to evaluate the performance of our simple estimator on model selection for the random graph generator in Section 5.3.

- **DARTS**: we produce this dataset by randomly sampling 100 architectures from the search space used in DARTS (Liu et al., 2019) and evaluating them on CIFAR10. This search space is more general than that of NASBench-201 and widely adopted in NAS (Zoph et al., 2018; Liu et al., 2019; Chen et al., 2019; Xie et al., 2019b; Xu et al., 2019; Real et al., 2019; Li & Talwalkar, 2020; Pham et al., 2018; Shaw et al., 2019; Zhou et al., 2020). We experiment with different evaluation set-ups in Section 5.2 and use this dataset to assess the stability/robustness of our estimator as well as make comparison to **TLmini** and **VAcc(EMA)**.

More details on the three datasets are provided in Appendix A. In NAS, the relative performance ranking among different models matters more than the exact test performance of models. Thus, we evaluate different performance estimators by comparing their rank correlation with the model's true/final test accuracy. We adopt Spearman's rank correlation following (Ying et al., 2019; Dong & Yang, 2020). We flip the sign of SoTL/SoTL-E/SoVL/TLmini (which we want to minimise) to compare to the Spearman's rank correlation of the other methods (which we want to maximise). We test different summation window sizes in Appendix C and find $E = 1$ consistently give the best results. Thus, we set $E = 1$ as the default choice for our SoTL-E estimator in the following experiments. Note SoTL-E with $E = 1$ corresponds to the sum of training losses over all the batches in one single epoch. All experiments were conducted on a 36-core 2.3GHz Intel Xeon processor with 512 GB RAM.

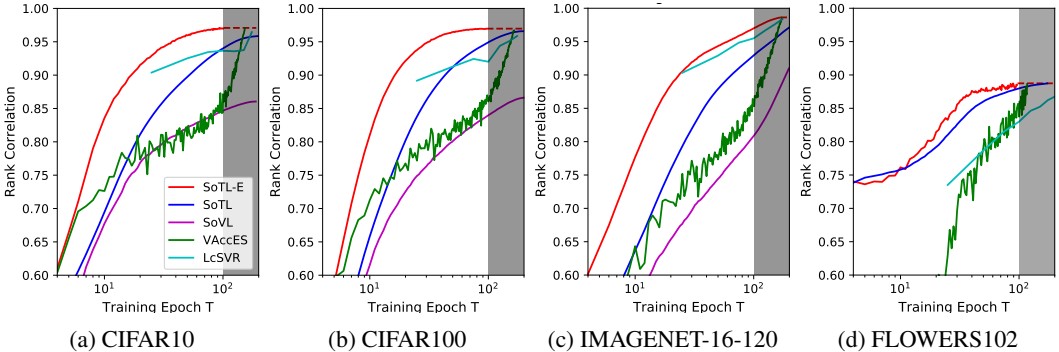

(a) CIFAR10      (b) CIFAR100      (c) IMAGENET-16-120      (d) FLOWERS102

Figure 2: Rank correlation performance of various baselines: SoTL-E, SoTL, SoVL, Val Acc and LcSVR for 5000 random architectures in NASBench-201 on three image datasets (a) to (c) and for 552 randomly wired architectures on FLOWERS102 (d). In all cases, our SoTL-E achieves superior rank correlation with the true test performance in much fewer epochs than other baselines. We shade the region $T > 100$; this shaded region is less interesting in NAS where we want to use as fewer training epochs as possible to maximise the speed-up gain compared to full evaluation $T = T_{end}$.

## 5.1 TRAINING LOSS VS VALIDATION LOSS

We perform a simple sanity check against the validation loss on NASBench-201 datasets. Specifically, we compare our proposed estimators, SoTL and SoTL-E, against two equivalent variants of validation loss-based estimators: SoVL and Sum of validation losses over the most recent epoch (SoVL-E with $E = 1$). For each image dataset, we randomly sample 5000 different neural network architectures from the search space and compute the rank correlation between the true test accuracies (at $T = 200$) of these architectures and their corresponding SoTL/SoTL-E as well as SoVL/SoVL-E up to epoch $T$. The results in Fig. 1 show that our proposed estimators SoTL and SoTL-E clearly outperform their validation counterparts.

Another intriguing observation is that the rank correlation performance of SoVL-E drops significantly in the later phase of the training (after around 100 epochs for CIFAR10 and 150 epochs for CIFAR100) and the final test loss, TestL (T=200), also correlates poorly with final test *accuracy*. This implies that the validation/test losses can become unreliable indicator for the validation/test accuracy on certain datasets; as training proceeds, the validation accuracy keeps improving but the validation losses could stagnate at a relatively high level or even start to rise (Mukhoti et al., 2020; Soudry et al., 2018). This is because while the neural network can make more correct classifications on validation points (which depend on the argmax of the logits) over the training epochs, it also gets more and more confident on the correctly classified training data and thus the weight norm and maximum of the logits keeps increasing. This can make the network overconfident on the misclassified *validation* data and cause the corresponding validation loss to rise, thus offsetting or even outweighing the gain due to improved prediction performance (Soudry et al., 2018). Training loss won't suffer from this problem (Appendix D). While SoTL-E struggles to distinguish architectures once their training losses have converged to approximately zero, this contributes to a much smaller drop in estimation performance of SoTL-E compared to that of SoVL-E and only happens near the very late phase of training (after 150 epochs) which will hardly be reached if we want efficient NAS using as *few* training epochs as possible. Therefore, the possibility of network overconfidence under misclassification is another reason for our use of training losses instead of the validation losses.

## 5.2 COMPARISON AGAINST OTHER BASELINES

We now compare our estimators SoTL and SoTL-E against other baselines mentioned at the start of Section 5. The results on both NASBench-201 and RandWiredNN datasets are shown in Fig. 2. Our proposed estimator SoTL-E, despite its simple form and cheap computation, outperforms all other methods under evaluation for $T < 100$ for all architecture/image datasets. Although the validation accuracy(VAccES) at $T \geq 150$ can reach similar rank correlation, this is less interesting for applications like NAS where we want to speed up the evaluation as much as possible and thus use as fewer training epochs as possible. The learning curve extrapolation method, LcSVR, is competitive.

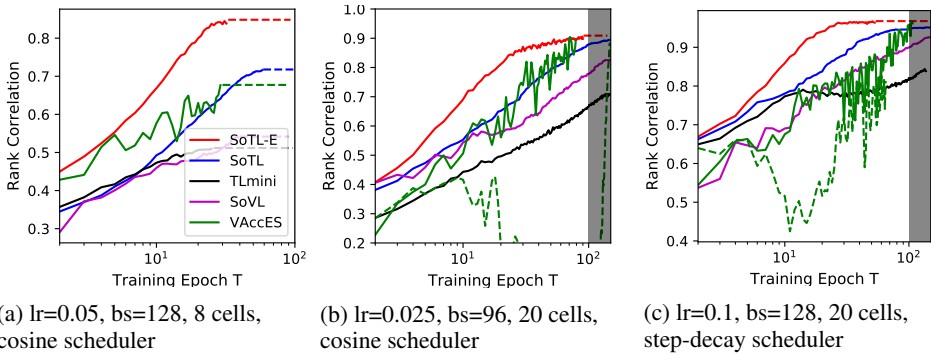

(a) lr=0.05, bs=128, 8 cells, cosine scheduler

(b) lr=0.025, bs=96, 20 cells, cosine scheduler

(c) lr=0.1, bs=128, 20 cells, step-decay scheduler

Figure 3: Rank correlation performance of various baselines: SoTL-E, SoTL, SoVL, VAccES, VAccES(EMA) and TLmini for 100 random architectures from DARTS search space on CIFAR10. We test the three training set-ups used in (Liu et al., 2019): (a) the search phase (over 200 architectures), (b) the retraining phase for CIFAR10 and (c) the retraining phase for ImageNet. Our SoTL-E achieves superior rank correlation in much fewer epochs than other baselines. Note also that the sum of training loss (SoTL/SoTL-E) gives better test performance estimation than individual training loss (TLmini).

However, the method requires hundreds of fully trained architecture data[2] to train the regression surrogate. Lots of computational resources are needed to obtain such training data.

We further verify the robustness of our estimator across different training set-ups adopted in (Liu et al., 2019) on the DARTS dataset. Specifically, we evaluated on architectures of different sizes (8 cells and 20 cells) as well as different training set-ups (initial learning rate, learning rate scheduler and batch size). The results in Fig. 3 show that our estimator again outperforms the competing methods. Note here the curve of **TLmini** corresponds to the average rank correlation with final test accuracy achieved by the mini-batch training loss over the epoch. The clear performance gain of our SoTL estimator over TLmini supports our claim that it is the sum of training losses, which carries the theoretical interpretation explained in Section 3, instead of the training loss at a single minibatch, that serves as a good estimator of generalisation performance. Further, the results of VAcc(EMA) show that the EMA technique, which smooths and improves the accuracies during validation, does not necessarily improve the rank correlation of validation accuracy with the final test performance.

### 5.3 ARCHITECTURE GENERATOR SELECTION

For the RandWiredNN dataset, we use 69 different hyperparameter values for the random graph generator which generates the randomly wired neural architecture. Here we would like to investigate whether our estimator can be used in place of the true test accuracy to select among different hyperparameter values. For each graph generator hyperparameter value, we sample 8 neural architectures with different wiring. The mean and standard error of both the true test accuracies and SoTL-E scores over the 8 samples are presented in Fig. 4. Our estimator can well predict the relative performance ranking among different hyperparameters (Rank correlation$\geq 0.85$) based on as few as 10 epochs of training. The rank correlation between our estimator and the final test accuracy improves as we use the training loss in later epochs.

### 5.4 SPEED UP NAS

Similar to early stopping, our method is model-free and can significantly speed up the architecture performance evaluation by using information from early training epochs. In this section, we incorporate our estimator, SoTL-E, at $T = 50$ into several NAS search strategies: Regularised Evolution (Real et al., 2019) (top row in Fig. 5), TPE (Bergstra et al., 2011) (bottom row in Fig. 5) and Random Search (Bergstra & Bengio, 2012) (Appendix E) and performance architecture search on NASBench-201 datasets. We compare this against the other two benchmarks which use the final validation accuracy at $T = 200$, denoted as Val Acc (T=200) and the early-stop validation accuracy at $T = 50$, denoted as Val Acc (T=50), respectively to evaluate the architecture's generalisation

---

[2]We follow (Baker et al., 2017) and train the SVR on 200 architectures.

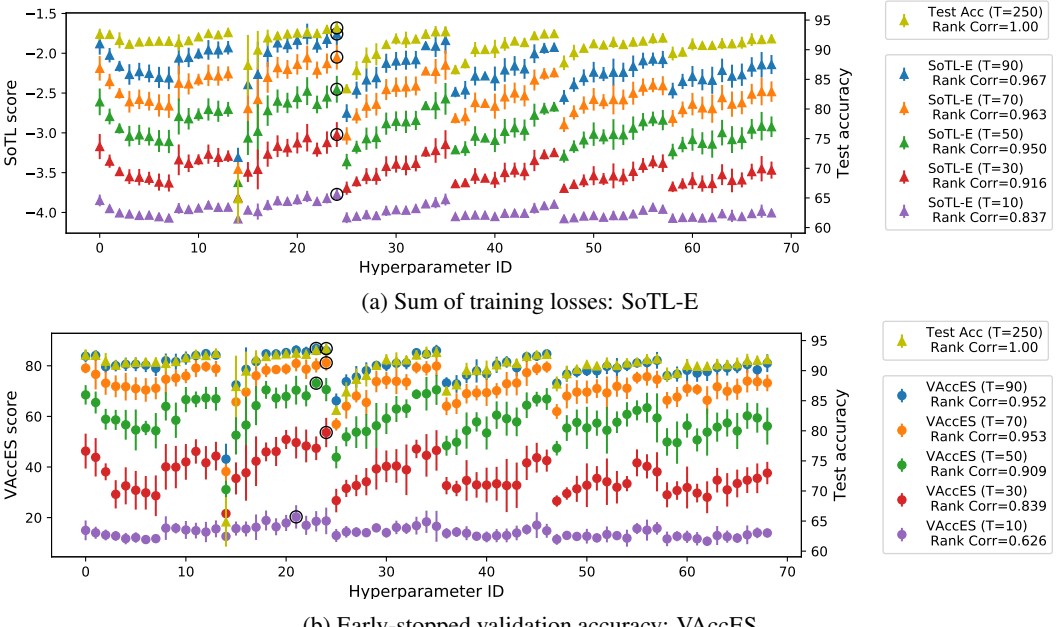

(a) Sum of training losses: SoTL-E

(b) Early-stopped validation accuracy: VAccES

Figure 4: Model selection among 69 random graph generator hyperparamters on RandWiredNN dataset using (a) our SOTL-E and (b) VAccES. We use each hyperparameter value to generate 8 architectures and evaluate their true test accuracies after complete training. The mean and standard error of the test performance across 8 architectures for each hyperparameter value are presented as Test Acc (yellow) and treated as ground truth (Right y-axis). We then compute our SoTL-E=1 estimator for all the architectures by using their first $T < 250$ epochs of training losses. The mean and standard error of SoTL-E scores for $T = 10, \dots, 90$ are presented in different colours (Left y-axis of (a)). The rank correlation between the mean Test Acc and that of SoTL-E for various $T$ is shown in the corresponding legends in (a). The same experiment is conducted by using early-stopped validation accuracy (VAccES) for performance estimation (b). With only 10 epochs of training, our SoTL-E estimator can already capture the trend of the true test performance of different hyperparameters relatively well (Rank correlation= 0.851) and can successfully identify 24-th hyperparamter setting as the optimal choice. The prediction of best hyperparameter by VAccES is less consistent and the rank correlation scores of VAccES at all epochs are lower than those of SoTL-E

performance. All the NAS search strategies start their search from 10 random initial data and are repeated for 20 seeds. The mean and standard error results over the search time are shown in Fig. 5. By using our estimator, the NAS search strategies can find architectures with lower test error given the same time budget or identify the top performing architectures using much less runtime as compared to using final or early-stopping validation accuracy. Also the gain of using our estimator is more significant for NAS methods performing both *exploitation* and exploration (RE and TPE) than that doing pure exploration (Random Search in Appendix E).

# 6 CONCLUSION

We propose a simple yet reliable method for estimating the generalisation performance of neural architectures based on its early training losses. Our estimator enables significant speed-up for performance estimation in NAS while outperforming other efficient estimators in terms of rank correlation with the true test performance. More importantly, our estimator has theoretical interpretation based on training speed and Bayesian marginal likelihood, both of which have strong links with generalisation. We believe our estimator can be a very useful tool for achieving efficient NAS.

## REFERENCES

Sanjeev Arora, Simon S Du, Wei Hu, Zhiyuan Li, and Ruosong Wang. Fine-grained analysis of optimization and generalization for overparameterized two-layer neural networks. *arXiv preprint*

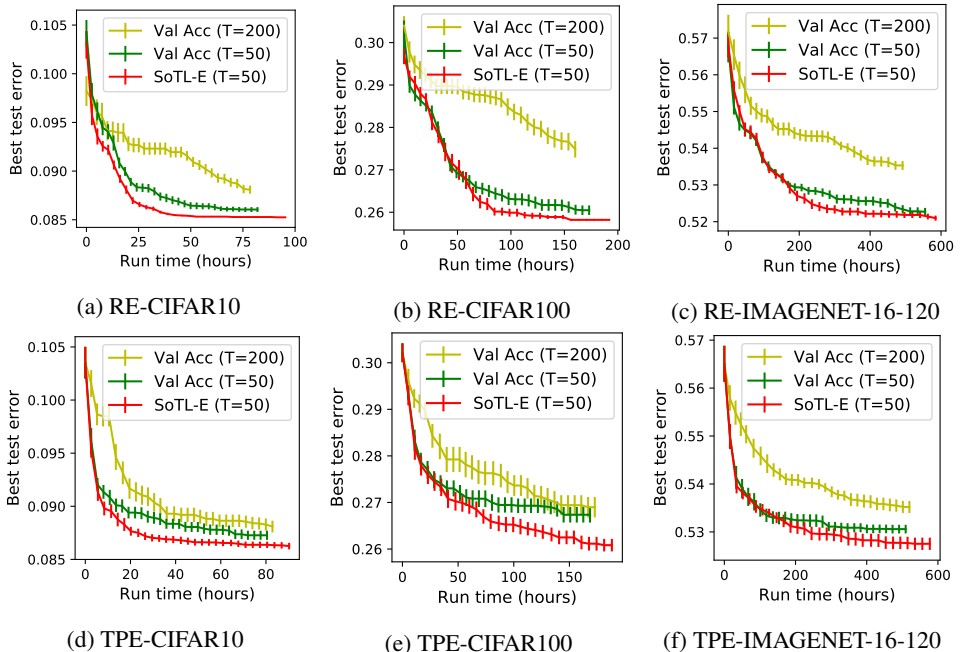

Figure 5: NAS performance of Regularised Evolution (RE) (Top row) and TPE (Bottom row) in combined with final validation accuracy (Val Acc (T=200)), early-stopping validation accuracy (Val Acc (T=50)) and our estimator SoTL-E on NASBench-201. SoTL-E leads to the fastest convergence to the top performing architectures in all cases.

*arXiv:1901.08584*, 2019.

Bowen Baker, Otkrist Gupta, Ramesh Raskar, and Nikhil Naik. Accelerating neural architecture search using performance prediction. *arXiv preprint arXiv:1705.10823*, 2017.

James Bergstra and Yoshua Bengio. Random search for hyper-parameter optimization. *Journal of machine learning research*, 13(Feb):281–305, 2012.

James S Bergstra, Rémi Bardenet, Yoshua Bengio, and Balázs Kégl. Algorithms for hyper-parameter optimization. In *Advances in Neural Information Processing Systems (NIPS)*, pp. 2546–2554, 2011.

Yuan Cao and Quanquan Gu. Generalization bounds of stochastic gradient descent for wide and deep neural networks. In H. Wallach, H. Larochelle, A. Beygelzimer, F. d' Alché-Buc, E. Fox, and R. Garnett (eds.), *Advances in Neural Information Processing Systems 32*, pp. 10836–10846. Curran Associates, Inc., 2019.

Xin Chen, Lingxi Xie, Jun Wu, and Qi Tian. Progressive differentiable architecture search. In *International Conference on Computer Vision (ICCV)*, 2019.

Tobias Domhan, Jost Tobias Springenberg, and Frank Hutter. Speeding up automatic hyperparameter optimization of deep neural networks by extrapolation of learning curves. In *Twenty-Fourth International Joint Conference on Artificial Intelligence*, 2015.

Xuanyi Dong and Yi Yang. Nas-bench-201: Extending the scope of reproducible neural architecture search. In *International Conference on Learning Representations*, 2020. URL `https://openreview.net/forum?id=HJxyZkBKDr`.

Gintare Karolina Dziugaite and Daniel M Roy. Computing nonvacuous generalization bounds for deep (stochastic) neural networks with many more parameters than training data. *arXiv preprint arXiv:1703.11008*, 2017.

Thomas Elsken, Jan Hendrik Metzen, and Frank Hutter. Neural architecture search: A survey. *arXiv:1808.05377*, 2018.

Stefan Falkner, Aaron Klein, and Frank Hutter. BOHB: Robust and efficient hyperparameter optimization at scale. In *International Conference on Machine Learning (ICML)*, pp. 1436–1445, 2018.

Moritz Hardt, Ben Recht, and Yoram Singer. Train faster, generalize better: Stability of stochastic gradient descent. In *International Conference on Machine Learning*, pp. 1225–1234, 2016.

Yiding Jiang*, Behnam Neyshabur*, Hossein Mobahi, Dilip Krishnan, and Samy Bengio. Fantastic generalization measures and where to find them. In *International Conference on Learning Representations*, 2020. URL https://openreview.net/forum?id=SJgIPJBFvH.

Nitish Shirish Keskar, Dheevatsa Mudigere, Jorge Nocedal, Mikhail Smelyanskiy, and Ping Tak Peter Tang. On large-batch training for deep learning: Generalization gap and sharp minima. *arXiv preprint arXiv:1609.04836*, 2016.

Aaron Klein, Stefan Falkner, Simon Bartels, Philipp Hennig, and Frank Hutter. Fast Bayesian optimization of machine learning hyperparameters on large datasets. *arXiv:1605.07079*, 2016a.

Aaron Klein, Stefan Falkner, Jost Tobias Springenberg, and Frank Hutter. Learning curve prediction with bayesian neural networks. 2016b.

Liam Li and Ameet Talwalkar. Random search and reproducibility for neural architecture search. *arXiv:1902.07638*, 2019.

Liam Li and Ameet Talwalkar. Random search and reproducibility for neural architecture search. In *Uncertainty in Artificial Intelligence*, pp. 367–377. PMLR, 2020.

Lisha Li, Kevin Jamieson, Giulia DeSalvo, Afshin Rostamizadeh, and Ameet Talwalkar. Hyperband: A novel bandit-based approach to hyperparameter optimization. *arXiv:1603.06560*, 2016.

Hanxiao Liu, Karen Simonyan, and Yiming Yang. DARTS: Differentiable architecture search. In *International Conference on Learning Representations (ICLR)*, 2019.

Tongliang Liu, Gábor Lugosi, Gergely Neu, and Dacheng Tao. Algorithmic stability and hypothesis complexity. In *Proceedings of the 34th International Conference on Machine Learning-Volume 70*, pp. 2159–2167. JMLR. org, 2017.

Clare Lyle, Lisa Schut, Binxin Ru, Mark van der Wilk, and Yarin Gal. A Bayesian perspective on training speed and model selection. *Thirty-fourth Conference on Neural Information Processing Systems*, 2020.

David JC MacKay. *Bayesian methods for adaptive models*. PhD thesis, California Institute of Technology, 1992.

David A McAllester. Pac-bayesian model averaging. In *Proceedings of the twelfth annual conference on Computational learning theory*, pp. 164–170, 1999.

Jishnu Mukhoti, Viveka Kulharia, Amartya Sanyal, Stuart Golodetz, Philip HS Torr, and Puneet K Dokania. Calibrating deep neural networks using focal loss. *arXiv preprint arXiv:2002.09437*, 2020.

Behnam Neyshabur, Srinadh Bhojanapalli, David McAllester, and Nati Srebro. Exploring generalization in deep learning. In *Advances in Neural Information Processing Systems*, pp. 5947–5956, 2017.

Maria-Elena Nilsback and Andrew Zisserman. Automated flower classification over a large number of classes. In *2008 Sixth Indian Conference on Computer Vision, Graphics & Image Processing*, pp. 722–729. IEEE, 2008.

Hieu Pham, Melody Guan, Barret Zoph, Quoc Le, and Jeff Dean. Efficient neural architecture search via parameter sharing. In *International Conference on Machine Learning (ICML)*, pp. 4092–4101, 2018.

Esteban Real, Sherry Moore, Andrew Selle, Saurabh Saxena, Yutaka Leon Suematsu, Jie Tan, Quoc V Le, and Alexey Kurakin. Large-scale evolution of image classifiers. In *International Conference on Machine Learning (ICML)*, pp. 2902–2911, 2017.

Esteban Real, Alok Aggarwal, Yanping Huang, and Quoc V Le. Regularized evolution for image classifier architecture search. In *Proceedings of the aaai conference on artificial intelligence*, volume 33, pp. 4780–4789, 2019.

Albert Shaw, Wei Wei, Weiyang Liu, Le Song, and Bo Dai. Meta architecture search. In *Advances in Neural Information Processing Systems*, pp. 11227–11237, 2019.

Daniel Soudry, Elad Hoffer, Mor Shpigel Nacson, Suriya Gunasekar, and Nathan Srebro. The implicit bias of gradient descent on separable data. *The Journal of Machine Learning Research*, 19 (1):2822–2878, 2018.

Mingxing Tan and Quoc V Le. Efficientnet: Rethinking model scaling for convolutional neural networks. *arXiv preprint arXiv:1905.11946*, 2019.

Saining Xie, Alexander Kirillov, Ross Girshick, and Kaiming He. Exploring randomly wired neural networks for image recognition. *arXiv:1904.01569*, 2019a.

Sirui Xie, Hehui Zheng, Chunxiao Liu, and Liang Lin. SNAS: Stochastic neural architecture search. In *International Conference on Learning Representations (ICLR)*, 2019b.

Yuhui Xu, Lingxi Xie, Xiaopeng Zhang, Xin Chen, Guo-Jun Qi, Qi Tian, and Hongkai Xiong. Pc-darts: Partial channel connections for memory-efficient differentiable architecture search. *arXiv preprint arXiv:1907.05737*, 2019.

Antoine Yang, Pedro M. Esperança, and Fabio M. Carlucci. NAS evaluation is frustratingly hard. In *International Conference on Learning Representations (ICLR)*, 2020.

Chris Ying, Aaron Klein, Eric Christiansen, Esteban Real, Kevin Murphy, and Frank Hutter. NAS-Bench-101: Towards reproducible neural architecture search. In *International Conference on Machine Learning (ICML)*, pp. 7105–7114, 2019.

Kaicheng Yu, Christian Sciuto, Martin Jaggi, Claudiu Musat, and Mathieu Salzmann. Evaluating the search phase of neural architecture search. In *International Conference on Learning Representations*, 2020. URL https://openreview.net/forum?id=H1loF2NFwr.

Arber Zela, Aaron Klein, Stefan Falkner, and Frank Hutter. Towards automated deep learning: Efficient joint neural architecture and hyperparameter search. *arXiv preprint arXiv:1807.06906*, 2018.

Arber Zela, Julien Siems, and Frank Hutter. Nas-bench-1shot1: Benchmarking and dissecting one-shot neural architecture search. In *International Conference on Learning Representations*, 2020. URL https://openreview.net/forum?id=SJx9ngStPH.

Chiyuan Zhang, Samy Bengio, Moritz Hardt, Benjamin Recht, and Oriol Vinyals. Understanding deep learning requires rethinking generalization. *arXiv preprint arXiv:1611.03530*, 2016.

Pan Zhou, Caiming Xiong, Richard Socher, and Steven CH Hoi. Theory-inspired path-regularized differential network architecture search. *arXiv preprint arXiv:2006.16537*, 2020.

Barret Zoph and Quoc Le. Neural architecture search with reinforcement learning. In *International Conference on Learning Representations (ICLR)*, 2017.

Barret Zoph, Vijay Vasudevan, Jonathon Shlens, and Quoc V Le. Learning transferable architectures for scalable image recognition. In *Proceedings of the IEEE conference on computer vision and pattern recognition*, pp. 8697–8710, 2018.

## A  DATASETS DESCRIPTION

The datasets we experiment with are:

- **NASBench-201** (Dong & Yang, 2020): the dataset contains information of 15,625 different neural architectures, each of which is trained with SGD optimiser and evaluated on 3 different datasets: CIFAR10, CIFA100, IMAGENET-16-120 for 3 random initialisation seeds. The training accuracy/loss, validation accuracy/loss after every training epoch as well as architecture meta-information such as number of parameters, and FLOPs are all accessible from the dataset. The search space of the NASBench-201 dataset is a 4-node cell and applicable to almost all up-to-date NAS algorithms. The dataset is available at `https://github.com/D-X-Y/NAS-Bench-201`.

- **RandWiredNN**: we produce this dataset by generating 552 randomly wired neural architectures from the random graph generators proposed in (Xie et al., 2019a) and evaluating their performance on the image dataset FLOWERS102 (Nilsback & Zisserman, 2008). We explore 69 sets of hyperparameter values for the random graph generators and for each set of hyperparameter values, we sample 8 randomly wired neural networks from the generator. A randomly wired neural network comprises 3 cells connected in sequence and each cell is a 32-node random graph. The wiring/connection within the graph is generated with one of the three classic random graph models in graph theory: Erdos-Renyi(ER), Barabasi-Albert(BA) and Watt-Strogatz(WS) models. Each random graph models have 1 or 2 hyperparameters which decide the generative distribution over edge/node connection in the graph. All the architectures are trained with SGD optimiser for 250 epochs and other training set-ups follow those in (Liu et al., 2019). This dataset allows us to evaluate the performance of our simple estimator on hyperparameter/model selection for the random graph generator. We will release this dataset after paper publication.

- **DARTS**: we produce this dataset by randomly sampling 100 architectures from the search space used in DARTS (Liu et al., 2019) and evaluating them on CIFAR10. This search space comprises a cell of 7 nodes. An architecture from this search space is formed by stacking the cell 8 or 20 times. Specifically, the first two nodes in cell $k$ are the input nodes which equals to the outputs of cell $k-2$ and cell $k-1$ respectively. The last node in the cell $k$ is the output node which gives a depthwise concatenation of all the intermediate nodes. The remaining four intermediate nodes are operation nodes take can take one out of eight operation choices. This search space is larger and more general than that of NASBench-201, and is also widely adopted in NAS (Zoph et al., 2018; Liu et al., 2019; Chen et al., 2019; Xie et al., 2019b; Xu et al., 2019; Real et al., 2019; Li & Talwalkar, 2020; Pham et al., 2018; Shaw et al., 2019; Zhou et al., 2020). In Section 5.2, we experiment with the three different evaluation set-ups used in (Liu et al., 2019):

  1. Search phase: We stack 8 cells to form the architecture and train the architecture for 150 epoch on CIFAR10 with a batch size of 128. We use the SGD optimiser with an initial learning rate of 0.05 and a cosine-annealing schedule, momentum of 0.9 and weight decay of $3 \times 10^{-4}$;

  2. Retraining phase for CIFAR10: We stack 20 cells to form the architecture and train the architecture for 150 epoch on CIFAR10 with a batch size of 96. We use the SGD optimiser with an initial learning rate of 0.025 and a cosine-annealing schedule, momentum of 0.9 and weight decay of $3 \times 10^{-4}$;

  3. Retraining phase for ImageNet: We stack 20 cells to form the architecture and train the architecture for 150 epoch on CIFAR10 with a batch size of 128. We use the SGD optimiser with an initial learning rate of 0.1 and a step-decay schedule (decayed by a factor of 0.97 after each epoch), momentum of 0.9 and weight decay of $3 \times 10^{-4}$.

  For this dataset, we also record the training loss for each minibatch and an alternative validation accuracy value evaluated using the exponential moving average (EMA) of the network weights (Tan & Le, 2019) on top of the conventional training and validation loss/accuracies. The minibatch training loss is used to verify our claim that it is the sum of training losses, which has nice theoretical interpretation, instead of individual training loss that gives good correlation with the generalisation performance of the architectures. The EMA version of the validation accuracy is used to check whether a smoothed and improved version of the early-stopped validation accuracy will have better correlation with the final true test performance.

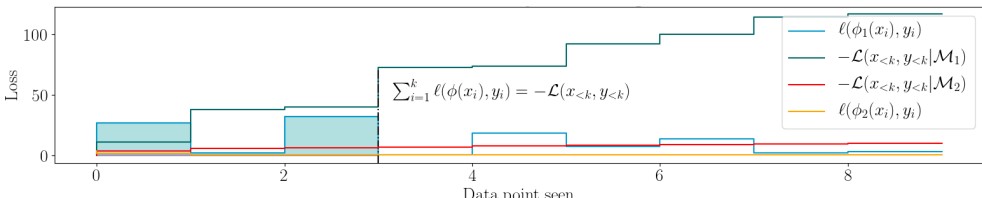

Figure 6: Example on a simple Bayesian linear regression problem. We see that the sum over training losses gives an estimator for the lower bound $\mathcal{L}$ of model evidence, and that the SoTL measure is more effective than the final training loss at distinguishing the two models $\mathcal{M}_1$ and $\mathcal{M}_2$.

## B    EXAMPLE ON BAYESIAN LINEAR REGRESSION

We illustrate how the SoTL metric corresponds to a lower bound on the marginal likelihood that can be used for model selection in a simple Bayesian linear regression setting. We consider an idealised data set $(X, y)$ with $X \in \mathbb{R}^{n \times (n+1)}$ and $y \in \mathbb{R}^n$, with $X$ of the form $X = (x_i)_{i=1}^n = ((y_i + \epsilon_0^i, 0, \ldots, \epsilon_i, \ldots, 0))_{i=1}^n$, and $\epsilon_i \sim \mathcal{N}(0, 1)$. We wish to compare two Bayesian linear regression models $\mathcal{M}_1$ and $\mathcal{M}_2$, each of which uses one of two different feature embeddings: $\phi_1$ and $\phi_2$, where $\phi_1(x) = x$ is the identity and $\phi_2(x) = x^\top e_1 = (y + \epsilon_0)$ retains only the single dimension that is correlated with the target, removing the noisy components of the input. The model which uses $\phi_2$ will have less opportunity to overfit to its training data, and will therefore generalise better than the model which uses $\phi_1$; similarly, it will also have a higher marginal likelihood. We demonstrate empirically in Fig. 6 that the SoTL estimator computed on the iterative posterior updates of the Bayesian linear regression models also exhibits this relative ranking, and illustrate how the SoTL relates to the lower bound described in Section 3.

## C    EFFECT OF SUMMATION WINDOW $E$

As shown in Fig. 1, summing the training losses over $E$ most recent epochs (SoTL-E) can achieve higher rank correlation with the true test accuracy than summing over all the previous $T$ epochs (SoTL), especially early on in training. We grid-search different summation window sizes $E = 1, 10, \ldots, 70$ to investigate the effect of $E$ and observe consistently across all 3 image datasets that smaller window size gives higher rank correlation during the early training phase and all $E$ values converge to the same maximum rank correlation (Fig. 7).

We further verify this observation by performing the same experiments on DARTS dataset for which we have saved the mini-batch training losses and thus can compute the sum of training losses for less than one epoch $E < 1$. For example, $E = 0.3$ corresponds to the sum of training losses over the first 30% of the mini-batches/optimisation steps in the epoch. The results in Fig. 8 show again that $E = 1$ is the optimal choices although smaller summation window in general leads to better performance than large window sizes at the very early part of the training. Thus, we recommend $E = 1$ as the default choice for our SoTL-E estimator. Note SoTL-E=1 corresponds to the sum of training losses over all the batches in one single epoch.

## D    TRAINING LOSSES VS VALIDATION LOSSES

### D.1    EXAMPLE SHOWING TRAINING LOSS IS BETTER CORRELATED WITH VALIDATION ACCURACY THAN VALIDATION LOSS

We sample three example architectures from the NASBench-201 dataset and plot their losses and validation accuracies on CIFAR100 over the training epochs $T$. The relative ranking for the validation accuracy is: Arch A (0.70) > Arch B (0.67) > Arch C (0.64), which corresponds perfectly (negatively) with the relatively ranking for the training loss: Arch A (0.05) < Arch B (0.31) < Arch C (0.69). Namely, the best performing architecture also has the lowest final training epoch loss. However, the ranking among their validation losses is poorly/wrongly correlated with that of validation accuracy; the worst-performing architecture has the lowest final validation losses but the

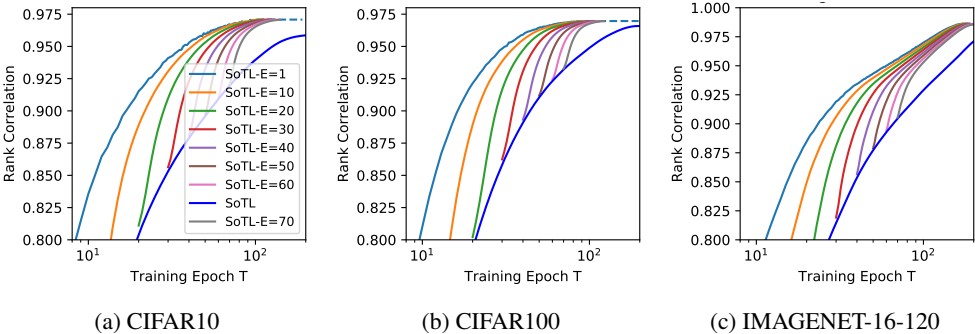

(a) CIFAR10                    (b) CIFAR100                    (c) IMAGENET-16-120

Figure 7: Rank correlation performance of the sum of training losses over $E$ most recent epochs (SoTL-E) on the NASBench-201 dataset. Different $E$ values are investigated for 5000 random architectures in NASBench-201 on three image datasets. In all three cases, smaller $E$ consistently achieves better rank correlation performance in the early training phase with $E = 1$ being the best choice.

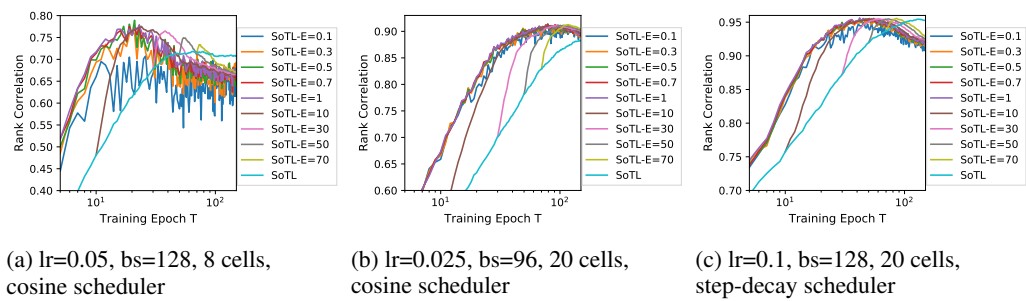

(a) lr=0.05, bs=128, 8 cells, cosine scheduler

(b) lr=0.025, bs=96, 20 cells, cosine scheduler

(c) lr=0.1, bs=128, 20 cells, step-decay scheduler

Figure 8: Rank correlation performance of the sum of training losses over $E$ most recent epochs (SoTL-E) on the DARTS dataset. Different $E$ values include those $< 1$ are investigated for 100 random architectures in DARTS search space under three different evaluation set-ups. In all three settings, smaller $E$ in general achieves better rank correlation performance in the early training phase with $E = 1$ again being the best choice. The performance of $E < 1$ is not stable and deteriorates from $E = 1$.

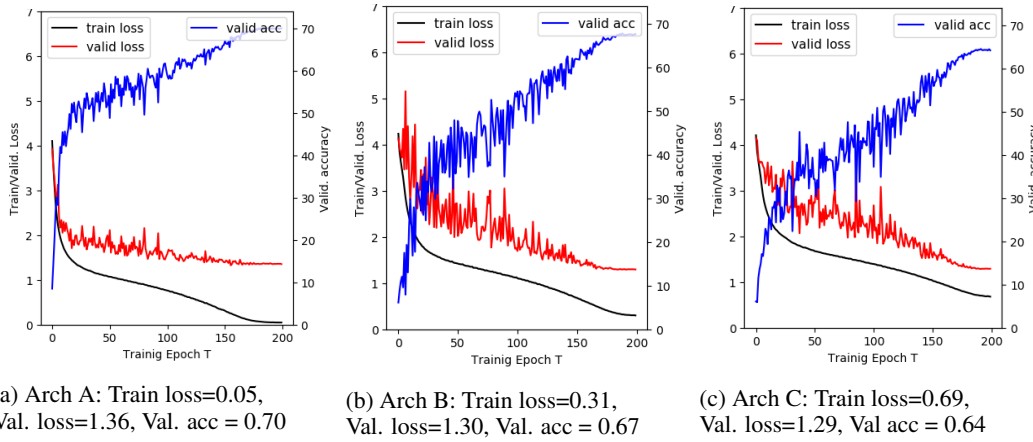

(a) Arch A: Train loss=0.05, Val. loss=1.36, Val. acc = 0.70

(b) Arch B: Train loss=0.31, Val. loss=1.30, Val. acc = 0.67

(c) Arch C: Train loss=0.69, Val. loss=1.29, Val acc = 0.64

Figure 9: Training losses, validation losses and validation accuracies of three example architectures on CIFAR100. The average of the training losses, validation losses and validation accuracies over the final 10 epochs is presented in the subcaption of each architecture.

best-performing architecture has the highest validation losses. Moreover, in all three examples, especially the better-performing ones, the validation loss stagnates at a relatively high value while the

validation accuracy continues to rise. The training loss doesn't have this problem and it decreases while the validation accuracy increases. This confirms the observation we made in Section 5.2 that the validation loss will become an unreliable predictor for the final validation accuracy as well as the generalisation performance of the architecture as the training proceeds due to overconfident misclassification.

## D.2    COMPARISON WITH SUM OVER VALIDATION ACCURACY

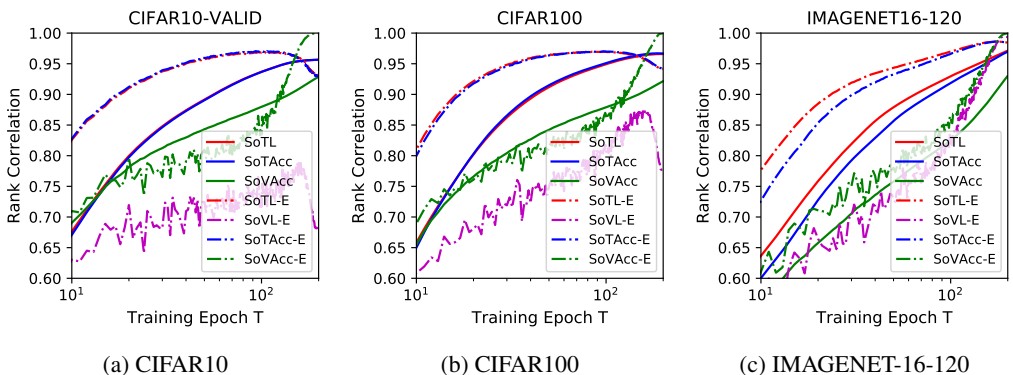

(a) CIFAR10        (b) CIFAR100        (c) IMAGENET-16-120

Figure 10: Rank correlation performance of the sum over training losses, SoTL (red), the sum over training accuracy, SoTAcc (blue), the sum over validation losses, SoVL (purple) and the sum of validation accuracy, SoVAcc (green) for 5000 random architectures in NASBench-201 on three image datasets. Note SoTL denotes the summation from epoch 0 to epoch T and SoTL-E denotes the summation over the most recent epoch T. The same applies for those of SoVL-E, SoTAcc/SoTAcc-E and SoVAcc/SoVAcc-E. The results on CIFAR10 and CIFAR100 confirm the discussion in Section 5.2 and in the subsection above; as the training proceeds, the validation loss can become poorly correlated with the validation/test accuracy while the training loss is still perfectly correlated with the training accuracy. Thus, another baseline to check against is the sum over validation accuracy, SoVAcc/SoVAcc-E. It's expected that SoVAcc-E should converge to a perfect rank correlation (=1) with the true test performance at the end of the training. However, the results in (a), (b) and (c) show that our proposed estimator *SoTL-E can consistently outperform SoVAcc-E* in the early and middle phase of the training (roughly $T \leq 150$ epochs). This reconfirms the usefulness of our estimator.

## D.3    OVERFITTING ON CIFAR10 AND CIFAR100

In Figure 2 in Section 5.2, the rank correlation achieved by SoTL-E on CIFAR10 and CIFAR100 will drop slighted after around $T = 150$ epochs but similar trend is not observed for IMAGENET-16-120. We hypothesise that this is due to the fact that many architectures converge to very small training losses on CIFAR10 and CIFAR100 in the later training phase, making it more difficult to distinguish these good architectures based on their later-epoch training losses. But this doesn't happen on IMAGENET-16-120 because it's a more challenging dataset. We test this by visualising the training loss curves of all 5000 architectures in Figure 11a where the solid line and error bar correspond to the mean and standard error respectively. We also plot out the number of architectures with training losses below 0.1 [3] in Figure 11b. It is evident that CIFAR10 and CIFAR100 both see an increasing number of overfitted architectures as the training proceeds whereas all architectures still have high training losses on IMAGENET-16-120 at end of the training $T = 200$ with none of them overfits. Thus, our hypothesis is confirmed. In addition, similar observation is also shared in (Jiang* et al., 2020) where the authors find the number of optimisation iterations required to reach loss equals 0.1 correlates well with generalisation but the number of iterations required going from loss equals 0.1 to loss equals 0.01 doesn't.

---

[3]the threshold 0.1 is chosen following the threshold for optimisation-based measures in (Jiang* et al., 2020)

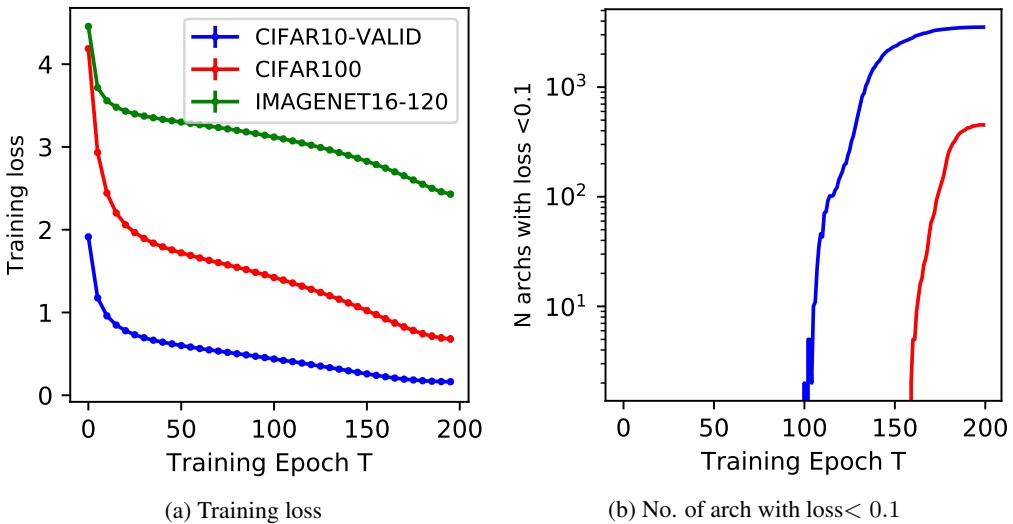

(a) Training loss

(b) No. of arch with loss$< 0.1$

Figure 11: Mean and 5 standard error of training losses and validation losses on 5000 architectures on different NASBench-201image datasets. (a) shows the training curves and (b) shows the number of architectures whose training losses go below 0.1 as the training proceeds. Many architectures reach very small training loss in the later phase of the training on CIFAR10 and CIFAR100, thus may overfitting on these two datasets. But all the architectures suffer high training losses on IMAGENET-16-120, which is a much more challenging classification task, and none of them overfits.

## E ADDITIONAL NAS EXPERIMENTS

In this work, we incorporate our estimator, SoTL-E, at $T = 50$ into three NAS search strategies: Regularised Evolution (Real et al., 2019), TPE (Bergstra et al., 2011) and Random Search (Bergstra & Bengio, 2012) and performance architecture search on NASBench-201 datasets. We modify the implementation available at `https://github.com/automl/nas_benchmarks` for these three methods.

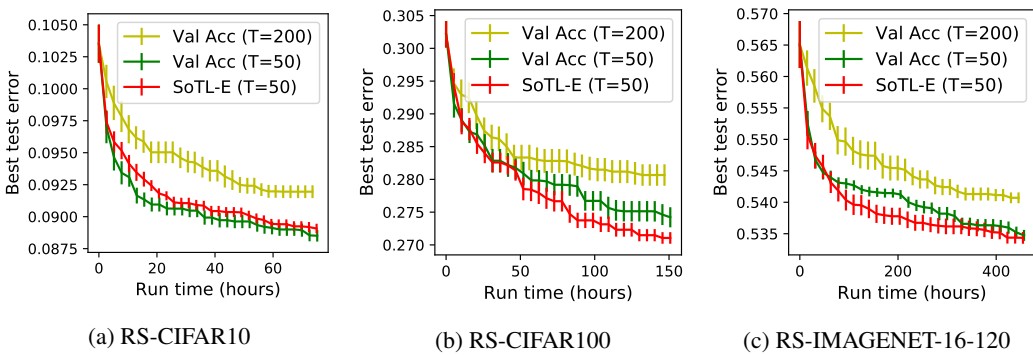

(a) RS-CIFAR10

(b) RS-CIFAR100

(c) RS-IMAGENET-16-120

Figure 12: NAS performance of Random Search (RS) in combined with final validation accuracy (Final Val Acc), early-stop validation accuracy (ES Val Acc) and our estimator SoTL-E on NASBench-201. SoTL-E enjoys competitive convergence as ES Val Acc and both are faster than using Final Val Acc.

Random Search (Bergstra & Bengio, 2012) is a very simple yet competitive NAS search strategy (Dong & Yang, 2020). We also combined our estimator, SoTL-E, at training epoch $T = 50$ with Random Search to perform NAS. We compare it against the baselines using the final validation accuracy at $T = 200$, denoted as Val Acc (T=200), and the early-stop validation accuracy at $T = 50$, denoted as Val Acc (T=50). Other experimental set-ups follow Section 5.5. The results over running

hours on all three image tasks are shown in Figure 12. The use of our estimator clearly leads to faster convergence as compared to the use of final validation i.e. Val Acc (T=200). Moreover, our estimator also outperforms the early-stop validation accuracy, Val Acc (T=50) on the two more challenging image tasks, CIFAR100 and IMAGENET-16-120, and is on par with it on CIFAR10. The performance gain of using our estimator or the early-stopped validation accuracy is relatively less significant in the case of Random Search compared to the cases of Regularised Evolution and TPE. For example, given a budget of 150 hours on CIFAR100, Regularised Evolution and TPE when combined with our estimator can find an architecture with a test error around or below 0.26 but Random Search only finds architecture with test error of around 0.27. This is due to the fact that Random Search is purely explorative while Regularised Evolution and TPE both trade off exploration and exploitation during their search; our estimator by efficiently estimating the final generalisation performance of the architectures will enable better exploitation. Therefore, we recommend the users to deploy our proposed estimator onto search strategies which involve some degree of exploitation to maximise the potential gain.

