# OpenReview forum: "Revisiting the Train Loss: an Efficient Performance Estimator for Neural Architecture Search"
_ICLR.cc/2021/Conference — Reject_

### Official Review · AnonReviewer4 · 2020-10-27
**Using training loss to improve generalization performance is unreasonable**

**Rating:** 3
**Confidence:** 3

**Review:**

###############################################################

Summary:

This paper provides a new method for estimating the generalization performance of neural architectures. This method used the sum of training loss as a criterion. The paper gave some intuitions about the method from the perspective of Bayeian model selection.

###############################################################

Reason for Score:

Using training loss to improve generalization performance is unreasonable. And the paper didn't give some convincing reason to this method. The method has no theoretical guarantee, and its analogy with Bayesian model selection seems problematic.

###############################################################

cons:

1, The method in this paper purely used training loss as a criterion for generalization performance. This is unreasonable. And the paper didn't give some convincing reason to this method.

2, The analogy with the Bayesian model selection is problematic. In Bayesian model selection, the parameter in the following step is the posterior estimator based on previous data. This paper think of the SGD optimizer as a way to find the posterior estimator. This is problematic.

---

### Official Review · AnonReviewer1 · 2020-10-28
**Review of Revisiting the Train Loss: an Efficient Performance Estimator for Neural Architecture Search**

**Rating:** 5
**Confidence:** 3

**Review:**

This paper proposes a simple model-free method to estimate the generalization performance of deep neural architectures based on their early training losses. The proposed method uses the sum of training losses during training to estimate the performance and is motivated by recent empirical and theoretical results. The experimental results show that the proposed estimator outperforms the existing methods that predict the performance ranking among architectures.

Pros
- The proposed approach is simple yet shows better performance than the existing estimator.
- This paper is discussed from both theoretical and empirical perspectives.

Cons
- I am wondering how the proposed estimator can be used in recent gradient-based NAS methods. For instance, DARTS optimizes architecture parameters during the search phase and pick up the top two operations that have the highest values after optimization. Is it possible to use the proposed estimator for DARTS optimization (i.e. recent gradient-based methods) and to speed up the optimization? Also, although One-shot NAS based on random sampling (e.g [1]) achieves good performance, can we apply the proposed method to such methods? I would like to know the scope of the application of the proposed method.
- In the first experiment, this paper randomly samples 100 architectures from the DARTS search space and evaluate the proposed estimator based on them. I would like to see the behavior of the proposed method with more samples to get a more accurate understanding because the search space of DARTS is much large.
- Regarding Figure 5, in the case of the regularized evolution (RE), I would like to know how much the speed of the optimization is improved. For instance, it would be nice to provide how much the proposed method can speed up the optimization to achieve the same performance reported in their paper. I am also interested in how fast it is compared with recent gradient-based NAS methods.
[1] G. Bender+, Understanding and Simplifying One-Shot Architecture Search, ICML, 2018

Overall, this paper proposes a simple yet effective method to estimate the generalization performance among deep neural networks and it is motivated by both empirical and theoretical aspects. However, there are some unclear points to be clarified for the publication as described above.

---

> ### Author Response · Authors · 2020-11-19
> **Response to Reviewer 1**
>
> Many thanks for your comments and the following are our responses.
>
> 1. How the proposed estimator can be used in gradient-based/one-shot NAS methods (e.g. DARTS)?
>
> Please refer to our response point 2) to Reviewer 3.
>
>
> 2. The scope of the application of the proposed method.
>
> The general scope of applying our proposed estimator is mainly for query-based NAS strategies where each queried architecture is trained independently following the same protocol (i.e. data transformation, optimiser hyperparameters type and regularisation techniques are all prespecified and fixed) and all the weights in the network are updated by the same number of steps. Despite being computationally more expensive, query-based NAS methods are more robust against overfitting to local optima than one-shot methods and can be easily extended to multi-objective settings. As shown in Fig. 5, our cheap estimator will also help alleviate the computational bottleneck of query based methods.
>
> 3. The behaviour of the proposed method with more samples from DARTS search space
>
> Please refer to our response 5 a) to Reviewer 3.
>
> 4. How much the speed of the optimization is improved for RE?
>
> The speed-up gain by using our SoTL for searching on NAS-Bench-201 is shown in Fig. 5 (a) to (c). For example on CIFAR100 in Fig. 5 (a),  using SoTL-E at training epoch 50 (red) to estimated the architecture performance reduce the search time of RE taken to achieve 74% test accuracy by around 2/3 compared to the use of early-stopped validation accuracy (green) and much more than 2/3 compared to the case of using the end-of-training validation accuracy (yellow). Note from Table 5 of (Dong & Yang, 2020), given the same search time, RE can achieve a mean test accuracy of 71.84% on CIFAR100, significantly outperforming the gradient-based methods ( the best is GDAS: 70.70%, the second best is only 59.05% and DARTS-V2 is only 15% due to overfitting to skip connections) on NAS-Bench-201.

---

### Official Review · AnonReviewer3 · 2020-10-29
**Official Blind Review #3**

**Rating:** 6
**Confidence:** 4

**Review:**

Instead of using validation accuracy to determine the efficacy of a network,  this paper recommends to use Sum over Training Losses (SOTL). SOTL-E is a variant where the sum of training losses begins to be computed after the first E epochs.
They also designed an early stopping mechanism based on Baker et al's SVR where they extrapolate SOTL instead of validation accuracy.

Questions:
1. How can training loss be used to identify a good network? It should theoretically lead to overfitting and poor generalization. Going by this argument, if we apply any kind of regularization such as dropout or weight decay, the training loss would not be low  while the test accuracy might still improve.
It is surprising that SOTL-E is able to rank the networks better than TestL at 200 for cifar10 and cifar100. Why do you think this is the case?

2. DARTS Experiment:
   (a) In  Figure 3(a), how is DARTS search replicated using just 100 random architectures? As it uses a SuperNet, it requires all possible architectures possible with the chosen operations.
   (b) In  Figure 3(b) and 3(c), the final architecture needs to be trained for 600 epochs. So it is natural that the rank correlation of SOVL, SOVL-E etc is poor for the first 100 epochs.

3. What would be more interesting is,  DARTS currently they perform a bi-level optimization. So instead of the architecture parameters $\alpha$ trying to minimize the validation loss, can they also minimize the training loss (theoretically this should not generalize well)? If not this, can you devise a way to plug in SOTL in the bi-level optimization and choosing the best architecture? If SOTL performs well even in that case, then you could a stronger case.

4.  Training very deep networks is not easy and takes more than 100 epochs to obtain good accuracy. So your observation might be a side effect of that too.  As SOTL could be applied to any deep learning networks, can you also repeat the experiment by training 100 networks sampled from a smaller search space, such as mobile search space (mobilenet, squeezenet, shufflenet etc), that takes less than 80 epochs to finish training, to see if it still holds true? Then use this SOTL and SOTL-E to determine the best network. Also compare with the baselines.

5. In Figure 4, how do SOVL, SOVL-E and Validation accuracy fare? Please include those too in the plots.

6. What is the difference between the setup of 1 and 2 (a) to (c) apart from the fact that SOVL-E and TestL are not included in 2?

7. If SOTL-E is the average training loss of the final epoch, why not call it that? (I understand that it is still the sum of losses for all the batches but as it not across epochs it is misleading).

As this is paper is proposing something that is fundamentally opposite to what has been studied widely thus far, it requires a lot more scrutiny. I do not think we can accept it with just empirical results and the theoretical motivation currently provided.

_____________________________
_____________________________
Post Rebuttal:

Thank you for replying to all of my questions..

Plugging in your new metric to DARTS seems to be promising, especially if it alleviates the DARTS collapse problem. Given that the community is more interested in one-shot NAS algorithms, this might be worthwhile pursuing

From the new plot in Figure 4 and NAS experiment in Figure 5, it is evident that the sum of training loss is able to rank the networks more effectively in the first 50 epochs. So one could use SOTL-E for early stopping rather than validation accuracy. This would also be effective in hyperband where the architectures are discarded after training them for very few epochs.

---

> ### Author Response · Authors · 2020-11-19
> **Response to Reviewer 3 Part 1**
>
> Thanks for your valuable comments. The following are our responses.
>
> 1. How can training loss be used to identify a good network? It should lead to overfitting and poor generalization…  regularization such as dropout or weight decay will increase training loss but improve test accuracy.
>
> [Justification for our method] Our estimator is based on the **sum** of training losses (i.e. the area under the training loss curve) and not the final training loss. The **sum of training losses**, especially at the early phase of training, is a direct measure of the training speed which has been successfully verified in previous works to correlate well with generalisation (Hardt et al., 2016; Negrea et al., 2019; Jiang & Neyshabur et al., 2020). Specifically, Jiang & Neyshabur et al. (2020) found that "number of iterations required to reach cross-entropy loss of 0.1" is predictive of generalisation performance. This metric is the same as our proposed estimator, and the connection is made explicit in our paper.
>
> [Overfitting cases] We agree that our estimator will fail to compare architecture performance when all architectures overfit and thus achieve near-zero training losses. But such a problem can only occur during the very **late phase of the training (near the end of the training)**, for which we also give evidence in Appendix D.3. In NAS settings, we instead focus on the **early phase of training** because we want to estimate the architecture performance using as few training epochs as possible to maximise the speed-up gain/cost-saving compared to running full evaluation (i.e. fully training the architecture for a large number of epochs). In such settings the models would have to overfit within the first few epochs of training for our method to fail.
>
> [Cheap estimation with small training budgets] Just to clarify, we don’t claim that our estimator can replace the gold standard of **final validation accuracy** in evaluating generalisation performance -- i.e. validation accuracy measured when we fully train the architecture. However, if you only have a small amount of training budget (tens of epochs) and want to have a decent estimation of the architecture’s final test accuracy, then we show that validation accuracy after a small number of epochs wouldn’t correlate well with final performance (VAccES in Fig. 2,3,4b), and we further show that our estimator achieves good correlation with final performance, and also outperforms other cheap alternatives (SoTL-E in Fig. 2,3,4a).
>
> [Problem setting] We’d like to clarify that we apply and investigate our estimator under the common NAS setting (adopted by **almost all NAS literatures** e.g. Zoph & Le, 2017; Real et al., 2019; Liu et al., 2019a; Liu et al., 2019b; Ying et al., 2019; White et al., 2019; Xu et al., 2020; Chen et al., 2020; Dong & Yang, 2020; Siems et al., 2020) where only the network architecture (i.e. type of operation units and the connection among these units in the network) is searched given the same training and evaluation protocol (i.e. data transformation, optimiser hyperparameters type and **regularisation techniques** are all prespecified and fixed). We don’t claim that our estimator can necessarily help distinguish the value of dropout or weight decay.
>
>
> 2. DARTS Experiment: run more random architectures for results in Fig. 3 ... in DARTS, the final architecture needs to be trained for 600 epochs so it is natural that the rank correlation of SOVL, SOVL-E etc is poor for the first 100 epochs.
>
> [More DARTS architectures] Given time and resource constraint, we only managed to sample and train 100 more random architectures from DARTS search space under the set-up in Fig. 3 (a) and update the Fig. 3 (a) accordingly. The conclusion and ranking between different methods remain the same and our proposed SoTL-E still outperforms other baselines.
>
> [Training epochs used] DARTS indeed uses 600 training epochs for complete evaluation. We set the training budget to 200 epochs for complete evaluation. However, we also adjust the learning rate scheduler to anneal the learning rate at a faster rate to accommodate the smaller training budget. Our approach is the same as the practice adopted in the NAS-Bench-301 dataset (Siems et al., 2020) where they reduce the training budget for each DARTS architecture to 100 epochs.
>
> (---Continued below---)

---

> > ### Author Response · Authors · 2020-11-19
> > **Response to Reviewer 3 Part 2**
> >
> > 3. Can the architecture parameters be optimised with training loss instead of the validation loss?.. can you plug in SOTL in the bi-level optimization to optimise the architecture?
> >
> > Several gradient-based one-shot NAS methods such as SNAS (Xie et al., 2018) , Single Path One-Shot NAS (Guo et al., 2020) and DS-NAS (Hu et al., 2020), which improves on DARTS, actually use the mini-batch training loss instead of mini-batch validation loss to optimise the architecture parameters. However, how much of the performance gain is due to the use of training loss remains unclear from those papers. Note the mini-batch training loss is different from our SoTL estimator as our estimator features the sum of mini-batch training losses.
> >
> > Following your suggestion, we tried naively replacing the validation loss in DARTS with our SoTL estimator (DARTS-SoTL) , and ran both DARTS-SoTL and original DARTS-ValLoss under the same setting for 4 seeds. The performance of the best architectures found over the seeds are shown below:
> >
> >   DARTS-SoTL: mean =  97.06 (std=0.25) , best=97.40
> >
> >   DARTS-VLoss : mean =  97.30 (std=0.07) , best=97.38
> >
> > In our paper, we've shown that SoTL is clearly better than SoVL so this experiment of using SoTL in place of validation loss doesn't support the superiority of our estimator but shows that in this setting they perform similarly. In fact, we didn’t expect our estimator to work directly in the weight-sharing/one-shot methods because their setup is quite different from what we investigated in the paper (which is for query-based NAS methods) ; our estimator is proposed for the case where each network is trained from scratch and all its weights are updated with **the same number of steps**. However, under the weight-sharing set-up, supernetwork weights that are more frequently selected and shared by the subnetwork candidates will get trained for more steps while those that are less shared will be trained for much fewer steps. Thus, the training loss of a subnetwork candidate is evaluated with weights trained for **different numbers of steps**. Better adapting our estimator to one-shot methods would be an interesting future direction.
> >
> > Another interesting observation is that the architecture cells found by DARTS-VLoss all have at least 1/4 and up to1/2 of the edge operations being skip connections while those by DARTS-SoTL contain none or only one skip connection. Thus, the use of SoTL may help alleviate the problem of DARTS overfitting to skip connections (Chen et al., 2019, Chu et al., 2019,  Liang et al., 2019, Zela et al., 2020, Dong & Yang, 2020).
> >
> >
> > 4. Why is SOTL-E able to rank the networks better than TestL at 200 for CIFAR tasks?
> >
> > We’ve explained this surprising phenomenon in Section 5.1 that the degraded correlation between final test loss and final test accuracy on CIFAR datasets was due to the network being overconfident on misclassified test data.
> >
> >
> > 5. Add SoVL-E and validation accuracy in Fig. 4
> >
> > We repeated the same experiments in original Fig. 4 for Val Acc and have added the results in the figure. It’s evident that our SoTL-E estimator enjoy higher rank correlation with the true test accuracy than Validation accuracy (VAccES) given the same training budget and can accurately predict the best generator hyperparameter at a much earlier epoch: epoch 10 for SoTL-E and epoch x for VAccES . We didn’t redo that for SoVL-E because we didn’t save the per-epoch validation loss when we generated the RandWiredNN-FLOWERS102 dataset previously and note that VAccES is always on par with, if not better than, SoVL and SoVL-E (Fig. 2 and Fig. 3)
> >
> >
> > 6. Difference between the setups of (a) to (c) in Fig. 1 and Fig. 2?
> >
> > The setup for Fig.1 and 2 are the same. We split them into 2 figures for clarity of presentation. Fig. 1 focuses on comparing test/validation loss against training loss to show the empirical advantage of using SoTL. Fig. 2 focuses on comparing a wider range of cheap performance estimators for NAS.

---

### Decision · Program_Chairs · 2021-01-07
**Final Decision**

**Decision:**

Reject

**Comment:**

The paper proposes to use the sum of training losses during training, or a variant where the sum of training losses begins to be computed after the first E epochs, to estimate the generalization performance of the corresponding network. Although the results seem promising for query-based NAS strategies, the reviewers agree that as the paper proposes something that is fundamentally opposite to the common practice, it requires more careful and thorough analysis. Besides, while the connection made by authors to the Bayesian marginal likelihood is interesting, it's not a rigorous argument that convinces the audience about the applicability of the proposed method. I strongly encourage the authors to add more analysis and discussion to the revised version to strengthen their claim and clarify its scope.